# Loop-Mediated Isothermal Amplification and Lateral Flow Immunochromatography Technology for Rapid Diagnosis of Influenza A/B

**DOI:** 10.3390/diagnostics14090967

**Published:** 2024-05-06

**Authors:** Woong Sik Jang, Jun Min Lee, Eunji Lee, Seoyeon Park, Chae Seung Lim

**Affiliations:** 1Emergency Medicine, College of Medicine, Korea University Guro Hospital, 148, Gurodong-ro, Guro-gu, Seoul 08308, Republic of Korea; plasmid18@korea.ac.kr; 2BK21 Graduate Program, Department of Biomedical Sciences, College of Medicine, Korea University, 145 Anam-ro, Seongbuk-gu, Seoul 02841, Republic of Korea; dlwnsals15@korea.ac.kr; 3Department of Laboratory Medicine, College of Medicine, Korea University Guro Hospital, 148, Gurodong-ro, Guro-gu, Seoul 08308, Republic of Korea; lucy5303@naver.com (E.L.); 08tjdus@naver.com (S.P.)

**Keywords:** influenza A, influenza B, loop-mediated isothermal amplification (LAMP), lateral flow assay (LFA)

## Abstract

Influenza viruses cause highly contagious respiratory diseases that cause millions of deaths worldwide. Rapid detection of influenza viruses is essential for accurate diagnosis and the initiation of appropriate treatment. We developed a loop-mediated isothermal amplification and lateral flow assay (LAMP-LFA) capable of simultaneously detecting influenza A and influenza B. Primer sets for influenza A and influenza B were designed to target conserved regions of segment 7 and the nucleoprotein gene, respectively. Optimized through various primer set ratios, the assay operated at 62 °C for 30 min. For a total of 243 (85 influenza A positive, 58 influenza B positive and 100 negative) nasopharyngeal swab samples, the performance of the influenza A/B multiplex LAMP-LFA was compared with that of the commercial Allplex^TM^ Respiratory Panel 1 assay (Seegene, Seoul, Korea). The influenza A/B multiplex LAMP-LFA demonstrated a specificity of 98% for the non-infected clinical samples, along with sensitivities of 94.1% for the influenza A clinical samples and 96.6% for the influenza B clinical samples, respectively. The influenza A/B multiplex LAMP-LFA showed high sensitivity and specificity, indicating that it is reliable for use in a low-resource environment.

## 1. Introduction

Influenza, commonly known as the flu, is a highly contagious respiratory illness caused by influenza viruses [1]. It poses a significant public health challenge globally, affecting millions of individuals annually [2]. The World Health Organization estimates that influenza results in about 3–5 million cases of severe illness and approximately 290,000–650,000 respiratory deaths worldwide each year [3]. The disease is particularly dangerous for high-risk groups such as the elderly, young children and individuals with underlying health conditions. Therefore, the swift and accurate diagnosis of influenza virus infections is pivotal for effective patient management and controlling outbreaks.

There are two primary types of influenza virus that cause seasonal epidemics in humans: influenza A and influenza B [4]. These viruses continually evolve, making their surveillance and detection a moving target for public health authorities [5]. Traditional methods for influenza virus detection, such as viral culture and serology, are time-consuming and often lack sensitivity [6]. Molecular techniques, notably reverse transcription-polymerase chain reaction (RT-PCR) techniques, have been acknowledged as the gold standard for detecting influenza due to their superior sensitivity and specificity. However, these methods are not without limitations, as they require complex infrastructures and specialized personnel, rendering them impractical in resource-constrained settings. This gap necessitates the development of a rapid, precise and cost-efficient diagnostic tool suitable for point-of-care application, particularly in low-resource environments [7,8,9].

Loop-mediated isothermal amplification (LAMP) assays have emerged as a viable alternative to RT-PCR, providing a simple, rapid and highly efficient amplification of nucleic acids operable at a constant temperature [10,11]. LAMP uses a unique set of 4–6 specially designed primers that bind to six distinct regions on the target gene, enabling the rapid and specific amplification of nucleic acid sequences under isothermal conditions, typically ranging from 60 to 65 °C. This method generates a large amount of DNA in a short period, with the process usually completing in less than an hour [12,13]. The amplification efficiency of LAMP, combined with its ability to operate without the need for thermal cycling, makes it particularly well suited for field settings and point-of-care applications [14]. Especially when coupled with lateral flow immunochromatographic assays (LFAs), LAMP assays can be transformed into a visually interpretable and user-friendly platform [15,16]. Specifically, the LAMP-LFA primers are designed to have two haptens (e.g., biotin and fluorescein isothiocyanate (FITC) or DIG) at their ends, with one hapten (e.g., biotin) bound to the interacting counterpart (e.g., streptavidin) immobilized on the LFA strip and the other hapten (e.g., FITC or DIG) bound to the matching antibodies (e.g., anti-FITC or anti-DIG antibodies) conjugated to gold nanoparticles. This approach allows for the simultaneous detection of multiple targets within a single analysis, significantly enhancing diagnostic efficiency and cost-effectiveness. Furthermore, the recent development of digital readers and smartphone-based applications for LFA reading technologies is extending from qualitative to semi-quantitative analysis, improving the precision and reliability of LFAs [17,18]. Therefore, the development of a multiplex LAMP-LFA enables the simultaneous detection of multiple targets within a single assay, significantly improving diagnostic efficiency and cost-effectiveness.

In this study, we develop a multiplex LAMP-LFA assay for influenza A and B, targeting the conserved regions within segment 7 of influenza A and the nucleoprotein gene of influenza B viruses. We assess the assay’s analytical and clinical performance by comparing it with the commercial Allplex™ Respiratory Panel 1 assay (Seegene, Seoul, Republic of Korea) by using analytical samples and 243 nasopharyngeal swab clinical samples, respectively.

## 2. Materials and Methods

### 2.1. Clinical Samples and RNA Extraction

Influenza A H1N1 and influenza B viruses were cultured at the Department of Laboratory Medicine in the Korea University Guro Hospital, of which the viral titers were measured using the TCID50 method. For clinical sensitivity testing, we used clinical influenza A nasopharyngeal (NP) swabs (*n* = 85), and influenza B NP (*n* = 58) samples collected from influenza A- and influenza B-afflicted patients (from February 2018 to July 2022) at Korea University Guro Hospital. These samples were stored at −70 °C until needed. All clinical samples were confirmed using the Anyplex II RV16 detection kit (Seegene, Inc., Seoul, Republic of Korea). To assess the specificity and cross-reactivity, 133 NP swab specimens from individuals without (100) and with (33) viral respiratory infections were tested. Respiratory viral infections, as confirmed via PCR using the Anyplex II RV16 detection kit, included nine coronaviruses (HKU1, NL63 and 229E), three SARS CoV-2 viruses, three respiratory syncytial A viruses (RSV As), three respiratory syncytial B viruses (RSV Bs), three adenoviruses (AdV), three parainfluenza virus (PIV) types 1–4, three human bocaviruses (HboVs), three human enteroviruses (HEVs), three human rhinoviruses (HRVs), and three metapneumoviruses (MPVs). Nucleic acids were extracted from all samples using Zentrix (Biozentech, Seoul, Republic of Korea) according to the manufacturer’s instructions. Briefly, 200 µL of the sample was dispensed into a 96 well extraction plate, and nucleic acid was extracted through the respiratory virus process program. This study was conducted in accordance with the guidelines of the Declaration of Helsinki and approved by the Institutional Review Board of Korea University Guro Hospital (approval number: 2021GR0546).

### 2.2. LAMP-LFA Primer Design

The influenza A and influenza B LAMP primer sets used in this study have been previously reported by our study group [19,20]. The primer sets for influenza A and influenza B were designed to target the conserved regions of segment 7 and the nucleoprotein, respectively. For the influenza A LAMP-LFA primer set, carboxyfluorescein (FAM) and biotin (BIO) were added to the 5’ end of Flu A_FIP and Flu A_BIP, respectively. For the influenza B LAMP-LFA primer set, digoxigenin (DIG) and biotin (BIO) were added to the 5’ end of Flu B_FIP and Flu B_FLP, respectively (Table 1). All LAMP-LFA primers and probes were synthesized by Macrogen Inc. (Seoul, Republic of Korea).

### 2.3. The Influenza A/B Multiplex LAMP-LFA

The influenza A/B multiplex LAMP-LFA consists of loop-mediated isothermal amplification (LAMP) to amplify the RNAs and lateral flow immunochromatography to detect the amplified RNAs. The RT-LAMP assay was performed using ELPIS RT-LAMP 2X Master Mix (Elpis-Biotech, Daejeon, Republic of Korea). For the influenza A/B multiplex LAMP-LFA, a reaction mixture was prepared with 12.5 μL of 2X Master Mix, 1 μL of influenza A LAMP-LFA primer mix, 1 μL of influenza B LAMP-LFA primer mix and 3 μL of sample RNA (final reaction volume: 25 μL). The mixture was placed on a heating block (Beijing HiYi Technology, Beijing, China) at 62 °C for 30 min. After the LAMP reaction, the lateral flow assay kit (Biozentech, Seoul, Republic of Korea) was placed on a clean, flat surface, and 5 μL of the LAMP product was added to 180 µL of LFA buffer. After 180 µL of the mixture was instilled into the sample well of the lateral flow assay kit, the results were analyzed with the naked eye and without equipment after 10 min. The influenza A/B multiplex LAMP-LFA had one control line and two test lines (T1, T2). When the influenza RNA samples were amplified, red bands appeared on the T1 and T2 lines for influenza A and influenza B, respectively.

### 2.4. The Allplex™ Respiratory Panel 1 Assay

The performance of the influenza A/B multiplex LAMP-LFA was compared with that of Allplex™ Respiratory Panel 1 using the CFX96 Touch Real-Time PCR Detection System (Bio-Rad, Hercules, CA, USA). Allplex™ Respiratory Panel 1 was used according to the manufacturer’s instructions. The PCR cycling conditions of Allplex™ Respiratory Panel 1 were as follows: reverse transcription at 50 °C for 20 min, inactivation at 95 °C for 15 min, 2 cycles of 95 °C for 10 s, 60 °C for 1 min and 72 °C for 10 s, followed by 45 cycles of 95 °C for 10 s, 60 °C for 1 min and 72 °C for 10 s with fluorescence detection at 60 °C and 72 °C.

### 2.5. Limit of Detection (LOD) Tests of the Influenza A/B Multiplex LAMP-LFA

To determine the detection limit of the influenza A/B multiplex LAMP-LFA, the extracted RNA from culture broth samples of influenza A H1N1 (2 × 10^6^ TCID50/mL) and influenza B (7 × 10^6^ TCID50/mL) were mixed with RNA isolated from a non-infected NP clinical sample. Each sample was serially diluted 10 fold from the original sample to obtain 6 levels. All tests were repeated 20 times and determined as the minimum concentration in a 10 fold dilution series at which 19 replicates were amplified.

## 3. Results

### 3.1. Optimization of the Influenza A/B Multiplex LAMP-LFA Primer Sets

For optimization of the influenza A/B multiplex LAMP-LFA, we evaluated the impact of different primer labeling strategies on the assay’s performance. Eight influenza A and B primer combinations were designed: two for the influenza A primer sets, labeled with biotin and FAM, and four for the influenza B primer sets, labeled with biotin and digoxigenin (DIG), as shown in Figure 1. The two influenza A primer sets featured different placements of biotin and FAM on the FIP and BIP. The four influenza B primer sets included different placements of biotin and DIG on the FIP and BIP, as well as the forward and backward loop primers (FLP and BLP, respectively). Out of the eight configurations, three particularly demonstrated clear and specific signaling without any cross-reactivity (Figure 1A,C,E). Among these, the most effective configuration was the one combining biotin-labeled FIP and FAM-labeled BIP for influenza A with biotin-labeled FIP and DIG-labeled FLP for influenza B (Figure 1A), which provided robust signals and high specificity.

### 3.2. Limit of Detection of the Influenza A/B Multiplex LAMP-LFA

The limit of detection (LOD) for the influenza A/B multiplex LAMP-LFA was compared with that of Allplex™ Respiratory Panel 1 using serial dilutions of influenza A H1N1 and influenza B RNA samples (Figure 2 and Table 2). For influenza A H1N1, the influenza A/B multiplex LAMP-LFA showed positive results at 2 × 10^2^ TCID50/mL, and Allplex™ Respiratory Panel 1 exhibited a limit of detection at 2 × 10^1^ TCID50/mL. In the case of influenza B, the influenza A/B multiplex LAMP-LFA showed positive detection at concentrations up to 7 × 10^3^ TCID50/mL, and Allplex™ Respiratory Panel 1 showed a limit of detection at 7 × 10^2^ TCID50/mL. For both influenza A and B, the influenza A/B multiplex LAMP-LFA displayed a limit of detection that was just one step higher than that of Allplex™ Respiratory Panel 1.

### 3.3. Comparison of Performance between the Influenza A/B Multiplex LAMP-LFA and Commercial Allplex™ Respiratory Panel 1

The clinical performance of the influenza A/B multiplex LAMP-LFA was compared with that of Allplex™ Respiratory Panel 1 using clinical samples (Table 3). For the samples from patients with influenza A (*n* = 85), Allplex™ Respiratory Panel 1 showed a sensitivity and specificity of 98.82% (95% CI: 93.99–99.95) and 100% (95% CI: 95.68–100), respectively. In comparison, the influenza A/B multiplex LAMP-LFA reported a sensitivity of 94.12% (95% CI: 86.53–97.75) with a specificity of 100% (95% CI: 95.48–100). In the case of influenza B (*n* = 58), Allplex™ Respiratory Panel 1 achieved a sensitivity and specificity of 100% (95% CI: 92.25–100). The influenza A/B multiplex LAMP-LFA showed a slightly lower sensitivity of 96.55% (95% CI: 86.28–99.48) and a specificity of 98.28% (95% CI: 89.14–99.92). For the negative control group of normal nasopharyngeal samples (*n* = 100), Allplex™ Respiratory Panel 1 correctly identified all samples as negative, yielding a specificity of 100% (95% CI: 95.48–100). The influenza A/B multiplex LAMP-LFA identified two false positives, both of which were cases of influenza A, resulting in a specificity of 98% (95% CI: 93.66–99.72). Overall, the influenza A/B multiplex LAMP-LFA showed acceptable sensitivity and specificity but had a slightly lower sensitivity and specificity than Allplex™ Respiratory Panel 1.

### 3.4. Cross-Reactivity Test of the Influenza A/B Multiplex LAMP-LFA

The influenza A/B multiplex LAMP-LFA was tested for cross-reactivity against a panel of viruses, including three samples each of SARS-CoV strain (229E, NL63 and OC43), SARS CoV-2, HEV, AdV, PIV1-4, MPV, HBoV, HRV, and RSV A and B (Table 4). No cross-reactivity was observed, with the assay producing zero positives across all tests for both influenza A and B. The influenza A/B multiplex LAMP-LFA assay demonstrated excellent specificity, showing no false detections among the common respiratory viruses tested, confirming its reliability for influenza diagnostics.

## 4. Discussion

Influenza is highly contagious worldwide and prevalent among high-risk groups, necessitating timely and accurate diagnosis [1,21]. Currently, the RT-qPCR test is considered the gold standard for its accuracy [22]. However, it is not suitable for field inspection due to the significant time, cost and specialized equipment required [23]. Recently, isothermal amplification methods, which operate at a constant temperature and are suitable for resource-limited environments or small medical facilities with limited access to expensive diagnostic devices or skilled technicians, have received much attention [9,24]. Numerous studies on isothermal amplification molecular diagnostics for influenza A and B are being conducted, utilizing methods such as loop-mediated isothermal amplification (LAMP) [25,26], recombinase polymerase amplification (RPA) [27] and helicase dependent amplification (HDR) [28]. Among these, the LAMP diagnostic method has undergone the most extensive research and has even been developed into commercial kits [29,30]. Much research has been carried out on multiplex diagnostic methods using LAMP with different types of fluorescent probes to diagnose more than one disease simultaneously [31,32]. However, the requirement for at least minimal fluorescence equipment presents a drawback for use in poorer countries or environments with limited resources [33].

The loop-mediated isothermal amplification-lateral flow assay (LAMP-LFA), which verifies amplified nucleic acids using an immunostrip, is suitable for on-site diagnostics. To date, several influenza LAMP-LFAs have been developed and reported [34,35,36]. This research has focused on developing singleplex LAMP-LFA methods, offering streamlined and user-friendly approaches for detecting the influenza virus. Recently, Akalına and Yazgan-Karataş took a significant step forward by developing a multiplex LAMP-LFA targeting both SARS-CoV-2 and influenza [37], broadening the scope of pathogens that can be detected simultaneously. Indeed, LAMP assays typically consist of 4–6 primers per target gene, and diagnosing more than two targets simultaneously requires at least 8–12 primers, potentially leading to a high incidence of non-specific reactions [38,39]. While multiplex LAMP assays using fluorescent probes often avoid frequent non-specific reactions due to precisely complementary target sequences [12], LAMP-LFAs risk frequent non-specific reactions due to dimer formation among primers targeting different diseases when tagged with biotin and FAM or DIG.

In this study, we developed an influenza A/B multiplex LAMP-LFA for the simultaneous diagnosis of influenza A and B. During optimization of the assay, we tested a total of eight sets of influenza A and B LAMP primer combinations, observing non-specific reactions in five of these. Out of the three remaining sets that showed no non-specific interactions, we selected the combination of the influenza A primer set (Flu A_FIP-Biotin and Flu A_BIP-FAM) and the influenza B primer set (Flu B_FIP-Biotin and Flu B_FLP-DIG) based on their superior signal strength. In clinical tests, the assay demonstrated sensitivities of 94.12% (95% CI: 86.53–97.75) for influenza A and 96.55% (95% CI: 86.28–99.48) for influenza B, with a specificity of 98% (95% CI: 93.66–99.72) for negative samples. Although the influenza A/B multiplex LAMP-LFA exhibited slightly lower sensitivity and specificity compared with the commercial Allplex™ Respiratory Panel 1, its sensitivity and specificity were still comparable. In addition, the assay’s lack of cross-reactivity with other respiratory viruses and its rapid turnaround time of approximately 30–40 min make it highly valuable in settings where quick diagnostics are critical or in environments with limited resources. From an economic perspective, the total cost per influenza A/B multiplex LAMP-LFA test is approximately USD 6–7, which includes reagents (USD 2–3), the lateral flow strip (USD 2) and other consumables (USD 2). This cost is significantly lower compared with the conventional RT-PCR methods, which typically range from USD 30 to 105 per test [40], excluding the need for sophisticated equipment and skilled personnel. Furthermore, the LAMP-LFA assay’s simplicity and the minimal requirement for equipment make it an economically viable option for rapid diagnostics in low-resource settings.

While the multiplex LAMP assay for influenza A/B developed in this study exhibits promising characteristics, it has limitations concerning practical field applications. All experiments in this study were conducted in a hooded environment that removed DNA and RNA contaminants, which does not accurately reflect actual field conditions. Therefore, aerosol-induced cross-contamination is a potential issue in real-world settings during the manual transfer and mixing of the LAMP product with LFA buffer, a critical step in the diagnostic process. Recently, various devices have been developed to facilitate safer and more reliable handling of bio-samples in non-laboratory settings [41,42,43]. These devices significantly reduce the risk of contamination by preventing aerosolization of the sample, thus maintaining the integrity of the assay. Therefore, integrating these devices with the influenza A and B LAMP assay will improve the applicability and reliability of the assay, making it a viable option for point-of-care testing in a variety of settings.

In summary, the influenza A/B multiplex LAMP-LFA developed in this study is a reliable and efficient diagnostic tool for detecting influenza A and influenza B. Its simplicity and quick turnaround time make it particularly useful for point-of-care testing in various settings.

## 5. Conclusions

The influenza A/B multiplex LAMP-LFA developed in this study, despite having a slightly higher detection limit compared with the commercial influenza diagnostic kit Allplex™ Respiratory Panel 1, demonstrated performance levels in terms of sensitivity and specificity that were comparable in clinical tests. This new assay notably showed the capability to accurately identify both influenza A and B viruses without cross-reactivity with other common respiratory viruses within a rapid processing time of approximately 30–45 min. This makes it a useful tool for timely diagnosis in clinical settings lacking a sophisticated laboratory infrastructure, and it is expected to be highly beneficial in environments where quick decision making is essential.

## Figures and Tables

**Figure 1 diagnostics-14-00967-f001:**
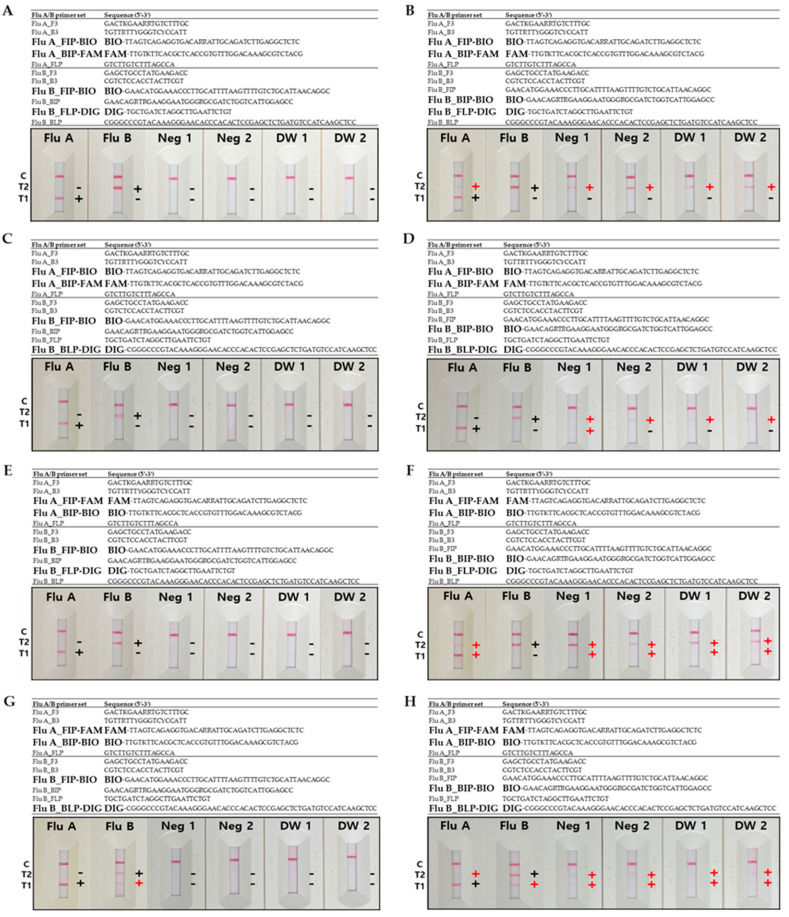
Optimization of the influenza A/B multiplex LAMP-LFA primer sets. (**A**) Combination of influenza A primer set (FIP-BIO and BIP-FAM) and influenza B primer set (FIP-BIO and FLP-DIG). (**B**) Combination of influenza A primer set (FIP-BIO and BIP-FAM) and influenza B primer set (BIP-BIO and FLP-DIG). (**C**) Combination of influenza A primer set (FIP-BIO and BIP-FAM) and influenza B primer set (FIP-BIO and BLP-DIG). (**D**) Combination of influenza A primer set (FIP-BIO and BIP-FAM) and influenza B primer set (BIP-BIO and FLP-DIG). (**E**) Combination of influenza A primer set (FIP-FAM and BIP-BIO) and influenza B primer set (FIP-BIO and FLP-DIG). (**F**) Combination of influenza A primer set (FIP-FAM and BIP-BIO) and influenza B primer set (BIP-BIO and FLP-DIG). (**G**) Combination of influenza A primer set (FIP-FAM and BIP-BIO) and influenza B primer set (FIP-BIO and BLP-DIG). (**H**) Combination of influenza A primer set (FIP-FAM and BIP-BIO) and influenza B primer set (BIP-BIO and FLP-DIG). C = control line; T1 = influenza A test line 1; T2 = influenza B test line 2; Flu A = influenza A; Flu B = influenza B; Neg = negative; DW = distilled water.

**Figure 2 diagnostics-14-00967-f002:**
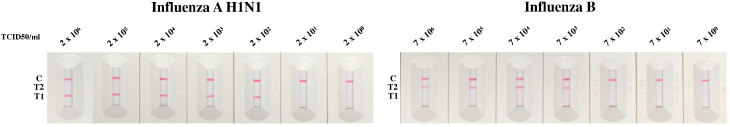
Limit of detection test of the influenza A/B multiplex LAMP-LFA for influenza A H1N1 and influenza B RNA samples. P = positive results; N = negative results; C = control line; T1 = influenza A test line 1; T2 = influenza B test line 2.

**Table 1 diagnostics-14-00967-t001:** The influenza A/B multiplex LAMP-LFA primers used in this study.

Target	Oligo	Sequence (5’-3’)	µM
Influenza A	Flu A_F3	GAC TKG AAR RTG TCT TTG C	2
	Flu A_B3	TGT TRT TYG GGT CYC CAT T	2
	Flu A_FIP-BIO	BIO-TTA GTC AGA GGT GAC ARR ATT GCA GAT CTT GAG GCT CTC	32
	Flu A_BIP-FAM	FAM-TTG TKT TCA CGC TCA CCG TGT TTG GAC AAA GCG TCT ACG	32
	Flu A_FLP	GTC TTG TCT TTA GCC A	8
Influenza B	Flu B_F3	GAG CTG CCT ATG AAG ACC	2
	Flu B_B3	CGT CTC CAC CTA CTT CGT	2
	Flu B_FIP-BIO	BIO-GAA CAT GGA AAC CCT TGC ATT TTA AGT TTT GTC TGC ATT AAC AGG C	32
	Flu B_BIP	GAA CAG RTR GAA GGA ATG GGR GCG ATC TGG TCA TTG GAG CC	32
	Flu B_FLP-DIG	DIG-TGC TGA TCT AGG CTT GAA TTC TGT	10
	Flu B_BLP	CGA GCT CTG ATG TCC ATC AAG CTC C	5

**Table 2 diagnostics-14-00967-t002:** Limit of detection test of Allplex™ Respiratory Panel 1 and influenza A/B multiplex LAMP-LFA for influenza A H1N1 and influenza B RNA samples.

Virus	TCID50/mL	Allplex™ Respiratory Panel 1				Influenza A/B Multiplex LAMP-LFA	
		Flu A	Flu B	RSV A/B	IC	Flu A	Flu B
		Cycle Threshold Values (Ct Values)				P/N	P/N
Influenza A H1N1	2 × 10^6^	19.76	N/A	N/A	26.98	P	N
	2 × 10^5^	24.19	N/A	N/A	26.12	P	N
	2 × 10^4^	27.92	N/A	N/A	25.92	P	N
	2 × 10^3^	32.53	N/A	N/A	25.32	P	N
	2 × 10^2^	36.66	N/A	N/A	25.65	P	N
	2 × 10^1^	39.83	N/A	N/A	25.48	N	N
	2 × 10^0^	N/A	N/A	N/A	25.35	N	N
Influenza B	7 × 10^6^	N/A	24.26	N/A	25.65	N	P
	7 × 10^5^	N/A	27.40	N/A	25.65	N	P
	7 × 10^4^	N/A	31.59	N/A	25.68	N	P
	7 × 10^3^	N/A	36.45	N/A	25.44	N	P
	7 × 10^2^	N/A	41.37	N/A	25.45	N	N
	7 × 10^1^	N/A	N/A	N/A	25.48	N	N
	7 × 10^0^	N/A	N/A	N/A	25.41	N	N

N/A = not applicable; P = positive results; N = negative results.

**Table 3 diagnostics-14-00967-t003:** Comparison of clinical performance of the influenza A/B multiplex LAMP-LFA and Allplex™ Respiratory Panel 1 for clinical samples.

Clinical Samples	Assay	P/N	Sensitivity (95% CI)	Specificity (95% CI)
Influenza A (*n* = 85)	Allplex™ Respiratory Panel 1	84/1	98.82 (93.63–99.79)	100 (95.68–100)
	Influenza A/B multiplex LAMP-LFA	80/5	94.12 (86.96–97.46)	100 (95.68–100)
Influenza B (*n* = 58)	Allplex™ Respiratory Panel 1	58/0	100 (93.79–100)	100 (93.79–100)
	Influenza A/B multiplex LAMP-LFA	56/2	96.55 (88.27–99.05)	98.28 (90.77–99.96)
Normal NP (*n* = 100)	Allplex™ Respiratory Panel 1	0/100	-	100 (96.3–100)
	Influenza A/B multiplex LAMP-LFA	2/98	-	98 (93–99.45)

P = positive results; N = negative results; CI = confidence interval.

**Table 4 diagnostics-14-00967-t004:** Cross-reactivity of the influenza A/B multiplex LAMP-LFA against other human infectious viruses.

Tested Clinical Samples	Influenza A/B Multiplex LAMP-LFA (Positive No./Test No.)	
	Influenza A	Influenza B
SARS CoV 229E	0/3	0/3
SARS CoV NL63	0/3	0/3
SARS CoV OC43	0/3	0/3
SARS CoV-2	0/3	0/3
HEV	0/3	0/3
AdV	0/3	0/3
PIV1	0/3	0/3
PIV2	0/3	0/3
PIV3	0/3	0/3
PIV4	0/3	0/3
MPV	0/3	0/3
HboV	0/3	0/3
HRV	0/3	0/3
RSV A	0/3	0/3
RSV B	0/3	0/3

CoV = coronavirus; HEV = human enterovirus; AdV = adenovirus; PIV = parainfluenza virus; MPV = metapneumovirus; HboV = human bocavirus; HRV = human rhinovirus; RSV A = respiratory syncytial virus A; RSV B = respiratory syncytial virus B.

## Data Availability

The raw data supporting the conclusions of this article will be made available by the authors on request.

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
