# Peer review of "Loop-Mediated Isothermal Amplification and Lateral Flow Immunochromatography Technology for Rapid Diagnosis of Influenza A/B"

_diagnostics, 2024, doi:10.3390/diagnostics14090967_

Round 1

Reviewer 1 Report

Comments and Suggestions for Authors

The authors developed a rapid assay for detection of influenza A/B infection using the combination of Loop-Mediated Isothermal Amplification a(LAMP) and lateral flow immuno-chromatography technology. Such method provides a rapid assay for diagnosis of influenza A/B infection with similar sensitivity as PCR.

Major comments:

How did the authors control cross-contamination when 180 µl of LFA buffer were added into 5 µl of LAMP-PCR products and 180 µl of the mixture were added into the lateral flow assay kit? Such steps were prone to cause aerosol by PCR products to surroundings.

 Other comments:

1.       Line 89, correct the world “KHU1 “ to “HKU1”

2.   No information about the storage of recruited clinical samples before testing was mentioned in “Materials and Methods” section.

3.     Figure 1, lack of information about “Combination 8 of Influenza A primer set and Influenza B primer set”

4.     Lines 119-120, there is discrepancy between 6 drops (180 µl ~ 200 µl) and 3 drops (180 µl).

5.  Line 202, which types of influenza were identified as false positive by LAMP-LFA?

6.    Why was SARS-CoV2 not be included in the panel for cross-reactivity test?

Author Response

Comments and Suggestions for Authors

The authors developed a rapid assay for detection of influenza A/B infection using the combination of Loop-Mediated Isothermal Amplification a(LAMP) and lateral flow immuno-chromatography technology. Such method provides a rapid assay for diagnosis of influenza A/B infection with similar sensitivity as PCR.

Response: Thank you for your thorough review and insightful comments on our manuscript titled " Loop-mediated isothermal amplification and lateral flow imunochromatography technology for rapid diagnosis of influenza A/B”. We have carefully considered each of the points raised and have made revisions accordingly.

Major comments:

How did the authors control cross-contamination when 180 µl of LFA buffer were added into 5 µl of LAMP-PCR products and 180 µl of the mixture were added into the lateral flow assay kit? Such steps were prone to cause aerosol by PCR products to surroundings.

Response: You are correct in noting the high potential for contamination when mixing the LAMP product with the buffer outside a controlled environment. We have addressed this issue in the Discussion section of our paper as follows:

“While the multiplex LAMP assay for Influenza A/B developed in this study exhibits promising characteristics, it has limitations concerning practical field applications. All experiments in this study were conducted in a hooded environment that removes DNA/RNA contaminants, which does not accurately reflect actual field conditions. Therefore, aerosol-induced cross-contamination is a potential issue in real-world settings during the manual transfer and mixing of the LAMP product with LFA buffer, a critical step in the diagnostic process. Recently, various devices have been developed to facilitate safer and more reliable handling of bio-samples in non-laboratory settings (42- 44). These devices significantly reduce the risk of contamination by preventing the aerosolization of the sample, thus maintaining the integrity of the assay. Therefore, integrating these devices with the influenza A/B LAMP assay will improve the applicability and reliability of the assay, making it a viable option for point-of-care testing in a variety of settings.” (Line 274-289)

 Other comments:

  1. Line 89, correct the world “KHU1 “ to “HKU1”

Response: We changed it. “KHU1 “ to “HKU1” (Line 90)

  1. No information about the storage of recruited clinical samples before testing was mentioned in “Materials and Methods” section.

Response: We have added the following at line 87 in the 'Materials and Methods' section: “These samples were maintained at -70°C until required for analysis.” (Line 85)

  1. Figure 1, lack of information about “Combination 8 of Influenza A primer set and Influenza B primer set”

Response: Following your suggestion, we have revised Figure 1 and updated the corresponding section in the results as follows:

“For the optimization of the influenza A/B multiplex LAMP-LFA, we evaluated the impact of different primer labeling strategies on the assay’s performance. Eight influenza A/B primer combinations were designed: two for influenza A primer sets, labeled with biotin and FAM, and four for influenza B primer sets, labeled with biotin and digoxigenin (DIG), as shown in Figure 1. The two influenza A primer sets featured different placements of biotin and FAM on the FIP and BIP primers. The four influenza B primer sets included different placements of biotin and DIG on the FIP and BIP primers, as well as the forward and backward loop primers (FLP and BLP). Out of the eight configurations, three particularly demonstrated clear and specific signaling without any cross-reactivity (Figure 1A, 1C, and 1E). Among these, the most effective configuration was the one combining biotin-labeled FIP and FAM-labeled BIP for Influenza A with biotin-labeled FIP and DIG-labeled FLP for Influenza B (Figure 1A), which provided robust signals and high specificity.” (Line 144-156)

  1. Lines 119-120, there is discrepancy between 6 drops (180 µl ~ 200 µl) and 3 drops (180 µl).

Response: We apologize for the discrepancy noted. To clarify, we used a pipette to mix 5 µl of LAMP product with 180 µl of buffer, and then 180 µl of this mixture was pipetted onto the immunostrip. The manuscript has been updated to reflect this precise methodology followed as: “5 μL of the LAMP product were added to 180µL of LFA buffer. After 180 µL of the mixture were instilled into the sample well of the lateral flow assay kit, the results were analyzed with the naked eye without equipment after 10 min.” (Line 120)

  1. Line 202, which types of influenza were identified as false positive by LAMP-LFA?

Response: To clarify which types of influenza were identified as false positives by the LAMP-LFA, we have specified in the manuscript that these included two instance both of influenza A, follows as: “The Influenza A/B multiplex LAMP-LFA identified 2 false positives, both of which were cases of influenza A, resulting in a specificity of 98% (95% CI: 93.66–99.72).” (Line 202)

  1. Why was SARS-CoV2 not be included in the panel for cross-reactivity test?

Response: Following your suggestion, we have tested the cross-reactivity of the Influenza A/B multiplex LAMP-LFA against three clinical SARS-CoV-2 samples. The details of this testing have been added to the 'Materials and Methods' (Line 90) and 'Results' (Line 212, Table 3) sections of the manuscript.

Reviewer 2 Report

Comments and Suggestions for Authors

This work introduces a LAMP-LFA with a high sensitivity and specificity, which is reliable for use in a low-resource environment. However, some important information is missing before it can be considered for publication. 

1. Figure 1 describes the optimization of the Influenza A/B multiplex LAMP–LFA primer sets. But the details of optimization is not provides, such important information should be provided, otherwise Figure 1 is meaningless. 

2. Whether a tool/device is suitable for point-of-care diagnostics is also related to the price of the tool or device. Such information is missing about this LAMP–LFA, I suggest the authors to add this information.

Comments on the Quality of English Language

1. The format of the title has a problem: the 'Rapid Diagnosis' should be 'rapid diagnosis'?

Author Response

Comments and Suggestions for Authors

This work introduces a LAMP-LFA with a high sensitivity and specificity, which is reliable for use in a low-resource environment. However, some important information is missing before it can be considered for publication.

Response: Thank you for your thorough review and insightful comments on our manuscript titled " Loop-mediated isothermal amplification and lateral flow imunochromatography technology for rapid diagnosis of influenza A/B”. We have carefully considered each of the points raised and have made revisions accordingly.

  1. Figure 1 describes the optimization of the Influenza A/B multiplex LAMP–LFA primer sets. But the details of optimization is not provides, such important information should be provided, otherwise Figure 1 is meaningless.

Response: Following your suggestion, we have revised Figure 1 and updated the corresponding section in the results as follows:

“For the optimization of the influenza A/B multiplex LAMP-LFA, we evaluated the impact of different primer labeling strategies on the assay’s performance. Eight influenza A/B primer combinations were designed: two for influenza A primer sets, labeled with biotin and FAM, and four for influenza B primer sets, labeled with biotin and digoxigenin (DIG), as shown in Figure 1. The two influenza A primer sets featured different placements of biotin and FAM on the FIP and BIP primers. The four influenza B primer sets included different placements of biotin and DIG on the FIP and BIP primers, as well as the forward and backward loop primers (FLP and BLP). Out of the eight configurations, three particularly demonstrated clear and specific signaling without any cross-reactivity (Figure 1A, 1C, and 1E). Among these, the most effective configuration was the one combining biotin-labeled FIP and FAM-labeled BIP for Influenza A with biotin-labeled FIP and DIG-labeled FLP for Influenza B (Figure 1A), which provided robust signals and high specificity.” (Line 144-156)

  1. Whether a tool/device is suitable for point-of-care diagnostics is also related to the price of the tool or device. Such information is missing about this LAMP–LFA, I suggest the authors to add this information.

Response: Based on your suggestion, we have added the following information about economic feasibility to the discussion section.

“From an economic perspective, the total cost per the Influenza A/B multiplex LAMP-LFA test is approximately $5-$7, which includes reagents, the lateral flow strip, and other consumables. This cost is significantly lower compared to the conventional RT-PCR methods, which typically range from $15-$25 per test, excluding the need for sophisticat-ed equipment and skilled personnel. Furthermore, the LAMP-LFA assay's simplicity and the minimal requirement for equipment make it an economically viable option for rapid diagnostics in low-resource settings.” (Line 267-273)

Comments on the Quality of English Language

  1. The format of the title has a problem: the 'Rapid Diagnosis' should be 'rapid diagnosis'?

Response: We have updated the title by changing 'Rapid Diagnosis' to 'rapid diagnosis' as suggested. (Line 3)

Round 2

Reviewer 2 Report

Comments and Suggestions for Authors

Good to see that the necessary information is provided now.